

# Evaluation of GPM IMERG Early, Late, and Final rainfall estimates with WegenerNet gauge data in southeast Austria

Sungmin O[1,2], Ulrich Foelsche[1,2,3], Gottfried Kirchengast[3,1,2], Jürgen Fuchsberger[3,1], Jackson Tan[4], and Walter A. Petersen[5]

[1]Institute for Geophysics, Astrophysics, and Meteorology/Institute of Physics (IGAM/IP), NAWI Graz, University of Graz, Austria
[2]FWF-DK Climate Change, University of Graz, Austria
[3]Wegener Center for Climate and Global Change (WEGC), University of Graz, Austria
[4]Universities Space Research Association, Columbia, Maryland, USA; NASA Goddard Space Flight Center, Greenbelt, Maryland, USA
[5]Earth Sciences Office, ST-11, NASA Marshall Space Flight Center, Huntsville, Alabama, USA

*Correspondence to:* Sungmin O (sungmin.o@uni-graz.at)

**Abstract.** The Global Precipitation Measurement (GPM) Integrated Multi-satellite Retrievals (IMERG) products provide quasi-global (60° N-60° S) precipitation estimates, beginning March 2014, from the combined use of passive microwave (PMW) and infrared (IR) satellites comprising the GPM constellation. The IMERG products are available in the form of near-real-time data, i.e. IMERG Early and Late, and of post-real-time research data, i.e. IMERG Final, after monthly rain gauge

analysis is received and taken into account. In this study, IMERG Early, Late, and Final (IMERG-E, -L, and -F) half-hourly rainfall estimates are compared with gauge-based gridded rainfall data from the WegenerNet Feldbach Region (WEGN) high density climate station network in southeast Austria. The comparison is conducted over two IMERG 0.1° x 0.1° grid cells, entirely covered by 40 and 39 WEGN stations each, with data during the extended summer season (April-October) for the first two years of the GPM mission. The entire data are divided into two rainfall intensity ranges (low and high) and two seasons

(warm and hot), and we evaluate the performance of IMERG using both statistical and graphical methods. Results show that IMERG-F rainfall estimates are in the best overall agreement with the WEGN data, followed by IMERG-L and IMERG-E estimates, particularly for the hot season. We also illustrate, through rainfall event cases, how insufficient PMW sources and errors in motion vectors can lead to wide discrepancies in the IMERG estimates. Finally, by applying the method of Villarini and Krajewski (2007), we find that IMERG-F half-hourly rainfall estimates can be regarded as a 25-min gauge accumulation,

with an offset of +40 min relative to its nominal time.

## 1 Introduction

The Global Precipitation Measurement (GPM) mission was launched in February 2014. This international mission is led by the National Aeronautics and Space Administration (NASA) and the Japan Aerospace and Exploration Agency (JAXA), as a successor to the Tropical Rainfall Measuring Mission (TRMM), to continue and improve satellite-based rainfall and snowfall

observations on a global scale (Tapiador et al., 2012; Hou et al., 2014; Yong et al., 2015). The GPM mission consists of a core





observatory satellite and a constellation of partner satellites to collect information from as many passive microwave (PMW) and infrared (IR) satellite platforms as available. Such a merged PMW-IR approach can mutually enhance the respective merits of individual PMW or IR satellite-based rainfall estimates, that is, IR satellite estimates can be adjusted with the greater accuracy of PMW data and, conversely, PMW satellite estimates can be interpolated along cloud movements obtained by the high

sampling rate of IR data (Kidd et al., 2003; Prigent, 2010; Kidd and Huffman, 2011; Kidd and Levizzani, 2011).

Once observation data are received from the PMW and IR platforms, they are combined into half-hourly gridded fields through the Integrated Multi-satellite Retrievals for GPM (IMERG) system (Huffman et al., 2015a, b). The IMERG system mainly comprises the following rainfall retrieval algorithms: the Climate Prediction Center Morphing-Kalman Filter (CMORPH-KF) (Joyce et al., 2004; Joyce and Xie, 2011), the Precipitation Estimation from Remotely Sensed Information

using Artificial Neural Networks (PERSIANN) (Sorooshian et al., 2000; Hong et al., 2004), and the TRMM Multi-Satellite Precipitation Analysis (TMPA) (Huffman et al., 2007). Processed differently based on user requirements in terms of data latency and accuracy, the IMERG computes Early, Late, and Final runs (hereafter IMERG-E, IMERG-L, and IMERG-F runs).

Since the first release of IMERG-F data in April 2014, extensive studies have been devoted for evaluation of the IMERG rainfall estimates against ground observations such as radars and gauges, or against other existing satellite rainfall data (e.g.

Guo et al., 2016; Liu, 2016; Prakash et al., 2016a, b; Sharifi et al., 2016; Tan et al., 2016; Tang et al., 2016). For instance, Tang et al. (2016) demonstrated, through an intercomparison study between the data using a hydrological model, that the IMERG products can adequately substitute TMPA products both statistically and hydrologically. Furthermore, Tan et al. (2016) presented a new validation approach for tracing rainfall errors to individual platforms or techniques within the IMERG system using ancillary variables provided in the products. Such analyses can provide useful information not only for further improve-

ments in processes of satellite rainfall retrieval but also for users in many relevant applications, from hydrological modeling and hazard studies to climate simulations (Barros et al., 2000; Nicholson et al., 2003; Bidwell et al., 2004; Wolff et al., 2005; Roca et al., 2010; Chen et al., 2013; Huang et al., 2013; Kirstetter et al., 2013; Lo Conti et al., 2014; Worqlul et al., 2014).

In this study, we evaluate and compare the rainfall data generated by all three IMERG runs using rain gauge-based gridded data from the WegenerNet Feldbach Region (WEGN) high density climate station network in southeast Austria (Kirchengast

et al., 2014). By this approach, the study aims to rigorously test the performance of IMERG runs and to explore differences between the data. The comparison is conducted over two 0.1° x 0.1° IMERG grid boxes, which are fully covered by 40 and 39 WEGN stations, respectively. We investigate the data during April-October in the years of 2014-2015; the first two years after the launch of the GPM Core Observatory. While IMERG-F data are available beginning April 2014, IMERG-E and IMERG-L data are available only from April 2015 at the time of this writing, so the evaluation of those two runs is restricted to 2015.

Note that all the IMERG data will eventually be retrospectively processed to the start of the TRMM era.

Even though gauge data are considered as ground reference in many existing validation studies, it is acknowledged that gauge measurements are also subject to uncertainties in terms of areal representativeness owing to a limitation in spatial coverage (Morrissey et al., 1995; Villarini et al., 2008). Fortunately, this point is of much less concern for the WEGN data. Around 40 gauges in one IMERG grid ensure a high reliability of data within the domain area, considering that a much smaller number

of gauges ranging from 5 to 15 gauges per 2.5° x 2.5° grid cell, depending on the study (Rudolf et al., 1994; Xie and Arkin,



1995; Ali et al., 2005; Villarini, 2010), has been suggested to guarantee a monthly error of under 10 %. Another concern is that tipping-bucket gauges, as employed in the WEGN network, involve systematic errors caused by various factors such as wind speed and rainfall intensity (Nešpor and Sevruk, 1999; Duchon and Essenberg, 2001). To this end, the WEGN data are adjusted by a correction factor described by O et al. (2016), who found that the WEGN tends to underestimate rainfall by about 10 %

compared to reference gauges.

The paper is organized as follows. Following this introduction, Sect. 2 further introduces IMERG and WEGN data and Sect. 3 describes the methodologies adopted for the assessment of IMERG estimates. The results are detailed in Sect. 4, in terms of statistical evaluation and analysis of example rainfall events. Section 5 contains concluding remarks and plans for future studies.

## 10 2   Data

### 2.1   GPM IMERG satellite rainfall estimates

IMERG version 3 (V03) level 3 products are used in this study. The level 3 products include gridded rainfall and snowfall data, with 0.1° x 0.1° spatial resolution and 30-min temporal resolution, generated from combining PMW and IR data of the GPM constellation satellites, and calibrated by gauge analysis of the Global Precipitation Climatology Centre (GPCC) (Schneider

et al., 2008). The IMERG processing steps include 1) the CMORPH-KF for quality-weighted time interpolation ("morphing") of PMW estimates following cloud motion vectors (Joyce et al., 2004; Joyce and Xie, 2011), 2) the PERSIANN-CCS for retrieving PMW-calibrated IR estimates (Sorooshian et al., 2000; Hong et al., 2004), and 3) the TMPA for inter-satellite calibration and monthly gauge adjustment (Huffman et al., 2007). A more complete data and algorithm description can be found in Huffman et al. (2015a).

The IMERG system is run twice in near-real-time (NRT), first to produce IMERG-E data about 6 h after nominal observation time for users who need a quick answer related to potential flood or landslides warning, and second to produce IMERG-L data with approximately 18 h latency for users working in agricultural forecasting or drought monitoring. Once the monthly gauge analysis is received, the final IMERG cycle is run to create the IMERG-F data approximately 3 months after the observation month. Note that both IMERG-E and IMERG-L runs use only part of the IMERG processing steps. For instance, instantaneous

PMW rainfall estimates are propagated only forward in time by the morphing scheme of the IMERG-E run, whereas both forward and backward morphing schemes are used in IMERG-L and IMERG-F runs. In this way, IMERG-L and IMERG-F runs are expected to better describe changes in the intensity and shape of rainfall features. For bias adjustment, the IMERG NRT runs use climatological gauge data, while the IMERG-F run ingests monthly GPCC gauge analyses, so the IMERG-F estimates are supposed to be most accurate and reliable (Huffman et al., 2015a, b). In this study, we use the calibrated estimates

(precipitationCal) for all IMERG runs.

IMERG version 4 (V04) products have been recently started to be released, but the new data do not yet cover the time ranges used in this study at the time of this writing (as of April 2017). The different version would not lead to significant changes in our conclusions, however, since the main aim of this study is to evaluate the three different IMERG runs relative to each other.





Moreover, preliminary intercomparison of IMERG V04 to V03 over limited time periods suggests no major change in overall performance from IMERG V03.

## 2.2   WEGN gridded rain gauge data

The WEGN is a high-resolution network for weather and climate study and monitoring purposes, located in the Feldbach region, southeast Austria (Kann et al., 2011; Kirchengast et al., 2014; Scheidl, 2014; Szeberényi, 2014; Kann et al., 2015). The region is part of the south-eastern Alpine foreland, characterized by the river Raab valley and a moderate hilly landscape with altitudes ranging from 260 to 600 m. The network comprises 153 weather stations in an area of about 300 km$^2$ (i.e., about one station per 2 km$^2$), collecting rainfall measurement data every 5 minutes (Fig. 1). 151 stations employ tipping-bucket gauges

for rainfall measurements and each gauge was equipped with one of three different sensors during the study period (Szeberényi, 2014). Meanwhile, since a major sensor replacement in 2016, all WEGN tipping-bucket gauges have employed the same type of sensor (O et al., 2016).

Once the WEGN processing system receives "Level 0" raw observations, with a latency of 1-1.5 hours, the Quality Control System produces "Level 1" station-level data. Then, only best quality Level 1 data are chosen to transfer into the Data Product

Generator (DPG), and the DPG generates the general user data products, "Level 2" station time series as well as 200 m x 200 m gridded data by an inverse-distance weighted interpolated method; all missing and non-best Level 1 data are filled in by temporal and spatial interpolation as part of the DPG processing. All data products are available online at the WEGN web portal within 2 hour latency. Since recently, based on the finding of O et al. (2016), the Level 2 processing also applies a bias correction factor for part of the rain data. More information on the WEGN data processing system and data products can be

found in Kabas et al. (2011) and Kirchengast et al. (2014).

For the statistical and rainfall event results reported in Sect. 4.1 and Sect. 4.2, half-hourly WEGN gridded rainfall data are used, which are generated by summing up the basic (5-min) gridded data, for direct comparison with IMERG rainfall estimates on a WEGN grid-points-average to IMERG grid box basis. For Sect. 4.3, on interpreting temporal characteristics of the satellite estimates, we use the 5-min gridded WEGN data in order to exploit this high native time resolution. For computing the area-

averaged WEGN rainfall for each IMERG grid box, we simply take the arithmetic mean of all WEGN grid points that lie within the grid box.

Furthermore, we use a threshold value of 0.05 to define rain/no-rain for the half-hourly data for preventing false alarms. Figure 2 shows that the number of WEGN half-hourly rainfall values retained after such threshold-clipping is significantly reduced only for a small number of gauges (less than about 30) and that more than about 65 gauges are not affected at all.

This suggests that the chosen threshold is reasonable, leaving a high amount of reliable half-hourly data, and that the WEGN half-hourly data exceeding 0.05 mm are very unlikely to be false alarms from the gauges' technical limitations. We also note that the WEGN is not a member network of the GPCC network so that the WEGN gauge data are independent from the IMERG gauge adjustment process.



## 3  Approach

We assess the performance of IMERG runs using both statistical and graphical methods. After inspecting some basic time series differences, we compare probability density functions (PDFs) and cumulative distribution functions (CDFs) of half-hourly IMERG estimates and WEGN data in terms of their distribution as function of rain rate. The PDF of rain occurrence (PDF$_c$) describes the percentages of rain-rate occurrence across the pre-defined bins. On the other hand, the CDF of rain volume (CDF$_v$) indicates the relative contribution of rain-rate in each bin to the total rain volume (Chen et al., 2013; Kirstetter et al., 2013). The PDFs and CDFs are computed over a binning range up to 30 mm, with a 0.5 mm bin width. We also use scatter plots to visually evaluate how IMERG estimates are distributed against the WEGN data.

In addition, we adopt widely used statistics and contingency indices, including Relative Bias (RB), Mean Absolute Error (MAE), Root Mean Squared Error (RMSE), Pearson Correlation Coefficient ($r$), Spearman's rank correlation coefficient ($\rho$), and Probability of Detection (POD), for quantifying differences in performance between the IMERG runs. These are used with definitions as follows.

$$RB = \frac{\sum_{i=1}^{n}(I_i - W_i)}{\sum_{i=1}^{n} W_i},$$ (1)

$$MAE = \frac{\sum_{i=1}^{n}|I_i - W_i|}{n}, \; and$$ (2)

$$RMSE = \sqrt{\frac{\sum_{i=1}^{n}(I_i - W_i)^2}{n}},$$ (3)

where $I_i$ and $W_i$ are respective rain rate values provided by IMERG estimates and WEGN data for a single grid box, at the $i$-th time step, with $n$ pairs of data. Related to the latter two, also Normalized MAE and RMSE (NMAE and NRMSE) are used, which are computed as relative values of MAE and RMSE with respect to the mean of the WEGN data. Furthermore,

$$r = \frac{cov(I,W)}{\sqrt{var(I)} \cdot \sqrt{var(W)}},$$ (4)

where $cov(X,Y)$ is the covariance between $X$ and $Y$ values, and $var(X)$ is the variance of $X$,

$$\rho = 1 - \frac{6\sum_{j=1}^{n}(rank_j(I) - rank_j(W))^2}{n(n^2 - 1)},$$ (5)

where $rank_j(X)$ means the rank position of $X$, and

$$POD = \frac{hits}{hits + misses}$$ (6)

where $hits$ means that both IMERG and WEGN data recorded rainfall ($\geq 0.05$ mm 30-min$^{-1}$), whereas $misses$ refers to the rainfall occurrences identified by WEGN data but missed by IMERG data. $POD$ ranges from 0 to 1 with a perfect score of 1





(in case of $misses = 0$).

Furthermore, we select two example rainfall events for case-based inspections of spatial patterns and time series, in order to visually explore some pronounced discrepancies, especially of IMERG-E and IMERG-L estimates, against WEGN data. The half-hourly WEGN gridded data from the whole network domain and corresponding time series of both IMERG estimates and

WEGN data are used in this evaluation with a consideration of data sources, i.e., PMW or IR observations.

Lastly, we employ the method of Villarini and Krajewski (2007) to provide an interpretation of IMERG rainfall estimates in terms of gauge accumulation time, $\Delta$, and the offset, $\delta$. WEGN 5-min gridded data are used as the basis for this purpose, test-integrating these gauge data in the range between 5 min and 100 min. The offset means the time from which to start the accumulation of gauge data and is considered to account for time differences between instantaneous satellite estimates and

actual rainfall on the ground surface. Consequently, we can reveal the combination of $\Delta$ and $\delta$ which leads to a minimum RMSE and interpret it as temporal resolution of the IMERG rainfall estimates.

When only the pairs for which both IMERG and WEGN data exceeding the threshold value are investigated, we generally classify the entire data into low rain intensities ($\leq 80^{\text{th}}$ percentile) versus high rain intensities ($> 80^{\text{th}}$ percentile), according to the percentiles of WEGN rain rates, and also into warm season (April, May, and October) versus hot season (June to

September), following the approach of Villarini and Krajewski (2007). We did not use data during the cold season, November to March, in order to guarantee the robustness of the WEGN data as ground reference, since most WEGN gauges are not heated and therefore do not capture snowfall events accurately (O et al., 2016).

## 4 Results

### 4.1 Statistical evaluation of IMERG rainfall estimates

Basic statistics of IMERG estimates and WEGN data are summarized in Table 1. All three IMERG estimates have a higher value for mean and maximum rain rates, and for standard deviation compared to those of WEGN data. The percentage of no-rain is also slightly larger for IMERG data, which is very likely related to limitations of satellite observations in detecting very low rain intensities (Kirstetter et al., 2012, 2013). Figure 3 shows the 24-h accumulated rainfall time series comparison (0.2 mm threshold is applied for the daily amounts). IMERG estimates and WEGN data show good overall agreement on the

occurrence of most daily rainfall events at IMERG grid scale, although IMERG tends to overestimate high rain rates.

Figure 4 shows $\text{PDF}_c$ and $\text{CDF}_v$ of IMERG estimates versus those of WEGN data. The IMERG estimates are in good agreement with the WEGN data in terms of rain occurrences except for low rain rates ($< 0.5 \, \text{mm} \, 30\text{-min}^{-1}$). However, IMERG shows high rain rates exceeding the maximum value of WEGN data ($15.3 \, \text{mm} \, 30\text{-min}^{-1}$), and consequently yields relatively large differences in $\text{CDF}_v$ against that of WEGN for the moderate to high rain rates. More specifically, rain rates less than

$15 \, \text{mm} \, 30\text{-min}^{-1}$ contribute to essentially 100 % of the total rain volume for WEGN data, while about 95 % for IMERG-F, and only about 75 % for IMERG NRT estimates. This shows that the satellite estimates sometimes overestimate rainfall and produce very high values, which have, in spite of their low frequency, a significant impact on the total rain volume.

Figure 5 is similar to Fig. 4 but with the data pairs restricted to those whose IMERG and WEGN values are both higher than





0.05 mm 30-min$^{-1}$, i.e., both detecting rain. Here, we also divided the entire data into low and high rainfall intensities, and into warm and hot seasons, as described in Sect. 3. Given that the disagreement of low rain rates in PDF$_c$ reduces (see entire data), it confirms that the differences seen in Fig. 4 are due to a poor sensitivity (misses) of satellites for low rain rates rather than due to some general biases in the estimation. Indeed, the $POD$ score of IMERG-E, IMERG-L, and IMERG-F estimates

is found 0.70, 0.79, and 0.75 against the WEGN data exceeding 0.5 mm 30-min$^{-1}$ (i.e., disregarding low rain rates), while it is only 0.50, 0.57, and 0.53 against the entire WEGN data.

In the panels of low rain intensities ($< 1.2$ mm 30-min$^{-1}$, or the $0.80^{\text{th}}$ percentile of WEGN data), IMERG CDF$_v$ still gets a contribution from rain rates greater than 1.2 mm 30-min$^{-1}$, even over 10 mm 30-min$^{-1}$, while the corresponding PDF$_c$ has a fairly good agreement with that of WEGN. This leads us to suspect that such big differences could be associated with

a time lag between rain peaks of IMERG estimates and WEGN data, rather than a tendency of the satellite to constantly overestimate rainfall; this will be further investigated and illustrated in Sect. 4.2. For the high intensities, the IMERG runs tend to underestimate the rain rates.

Furthermore, the CDF$_v$ of the IMERG NRT estimates does not show physically plausible shapes, e.g., a sudden rise between 10 and 20 mm. As seen in the hot season, the comparison reveals a clear improvement in IMERG rainfall estimates by applying

more retrieval or calibration processes to the satellite observations; CDF$_v$ moves closer to that of WEGN data, and also the shapes become gradually smoother from IMERG-E via IMERG-L to IMERG-F estimates. In general, it is concluded that IMERG-F estimates have the highest overall accuracy, followed by IMERG-L and IMERF-E estimates.

Figure 6 shows scatter plots of IMERG estimates versus WEGN data to enable a more quantitative understanding of the discrepancy between the data. Although it is a common practice to conduct regression analysis with scatter plots, we decided

not to because highly skewed distributions of rain rates (outliers) seen in the CDF$_v$ (Fig. 4) can strongly affect the results. Therefore, we chose to examine distributions of IMERG estimates over nine pre-defined rain rate bins (each bin containing at least 30 data pairs); $25^{\text{th}}$, $50^{\text{th}}$ (median), and $75^{\text{th}}$ percentiles of IMERG estimates are analyzed for each bin and the collective results shown in the panels of Fig. 6.

The IMERG runs show better performance (i.e., closer to one-to-one line) in estimating moderate rain rates within about

0.3 to 3 mm 30-min$^{-1}$, but a tendency to overestimate low rain rates and underestimate high rain rates. It is worth noting that the slopes of IMERG-F percentile lines (Fig. 6, top row) are consistent, with a relatively narrow spread, across the dataset partitioning, indicating that the biases in the IMERG-F estimates are relatively small and uniformly distributed. In contrast, the $75^{\text{th}}$ percentile line of the IMERG NRT estimates is slightly off the $50^{\text{th}}$ and $25^{\text{th}}$ percentile lines, particularly in the hot season, which indicates that the distribution of IMERG NRT rainfall estimates in the bins is skewed toward low values.

Table 2 provides the statistics metrics computed for each of the two IMERG grid boxes (see Fig. 1). All metrics are improved in IMERG-F estimates, except the correlation coefficient ($r$) that may not be a proper metric to evaluate the accuracy of IMERG data due to some large outliers. Indeed, IMERG-F estimates show the highest Spearman's rank correlation coefficient ($\rho$) which is known to be much less sensitive to outliers (Legates and McCabe, 1999; Habib et al., 2001). A somewhat better performance at Grid cell 15.85 compared to Grid cell 15.95 may be attributed to an Austrian national station within the WEGN area (over





Grid cell 15.85) of which measurements are integrated into the GPCC gauge product and which more influences Grid cell 15.85.

## 4.2    Analysis of example rainfall events

In this subsection, we focus on diagnosing the more detailed behavior of the IMERG runs by selecting example rainfall events

where the IMERG estimates show distinct differences from the WEGN data. Note that the WEGN can give very accurate information over the domain in terms of spatial and temporal rainfall variability, in spite of potential overall biases of up to about 10 % in the data. Figures 7 and  8 show the spatial distribution of two such rainfall events captured by the WEGN network and the corresponding time series of IMERG estimates and WEGN data.

Figure 7 shows a rainfall event in the warm season–30 May 2015. According to the spatial WEGN maps, the rain clouds

arrived at Grid cell 15.85 first (around 21:00 UTC), and then drifted eastwards. Among the IMERG NRT runs, the IMERG-L run is better able to describe this time lag between the two grids. This improvement can be attributed to backward morphing, which is applied in the IMERG-L, but not yet available in IMERG-E. The IMERG-L run captures rainfall withdrawal at Grid cell 15.95 better as well. Nevertheless, all IMERG runs tend to overestimate rainfall, with a time shift of about 2 hours earlier in starting time. This time shift suggests that the PMW observations (19:30-20:30 UTC) taken by the IMERG runs were likely

combined with incorrect IR cloud information. However, despite the absence of available PMW observations during the actual rainfall, the overestimation in IMERG-F is much smaller, thanks to the adjustment by the gauge analysis.

Figure 8 shows a rainfall event in the hot season–08 July 2015. Here again the onset of rainfall in IMERG estimates is ahead of that of the WEGN data (see the shaded area around 16:00 UTC). It is interesting that IMERG NRT runs well describe the first peak (13:30-14:00 UTC) at Grid cell 15.85, albeit with overestimation, but only the IMERG-E run captures the peak at

Grid cell 15.95 with a half-hour time shift. We suspect that the satellite-observed rain was morphed more slowly than the actual cloud movement (from west to east), for example, because the cloud motion vectors derived from IR-based data do not always accurately reflect the actual cloud advection speed (Joyce et al., 2004; Joyce and Xie, 2011), so the peak still remains at Grid cell 15.95 in the IMERG-E estimates.

When it comes to the IMERG-L and -F estimates, we assume that the backward morphing identified the timing of the peak

correctly. However, given that the morphing weights are inversely proportional to the time difference between the target data time and the PMW observation (i.e., higher weight is assigned for the time step when IMERG-E depicted the peak), the backward morphing significantly reduced the peak in the IMERG-E run (since it has a higher weight), whereas it only slightly increased the missing peak (since it has a lower weight). This implies a possibility of conflict between the forward and backward morphing that can lead to error in the rainfall estimates.

In Fig. 8, both IMERG-E and IMERG-L overestimate rainfall during 16:00-22:00 UTC. This can be explained by differences in the number of PMW observations involved in each IMERG run. The IMERG NRT runs could use four or fewer PMW observations during the period, all of which somehow overestimated the rain rates, resulting in the overestimation after the forward morphing and then even more after the backward morphing. According to Zeweldi and Gebremichael (2009), evaporation below cloud base can introduce large positive bias by the CMORPH morphing method during warm and hot seasons.





On the other hand, the IMERG-F run received more PMW-based information over the same period (see 18:00-18:30 UTC and 19:30-20:00 UTC) and the monthly gauge analysis. Thus, it shows better performance than the IMERG NRT runs. This demonstrates clearly the value of more PMW-based estimates in the morphing process (Joyce and Xie, 2011) as well as the ability of gauge adjustment to mitigate systematic biases. One reason the PMW observations overestimate rainfall is likely the
subgrid-scale rainfall variability. For instance, the IMERG runs may use satellite footprints over the northwestern corner of the grid cells, where rain is stronger ($\approx 15\ \mathrm{mm}\,30\text{-min}^{-1}$ at 16:30-17:00 UTC), for their gridding process.

### 4.3  Evaluation of temporal matching of IMERG estimates

Villarini and Krajewski (2007) used contour diagrams of RMSE as a function of accumulation time and time offset to interpret TMPA three-hourly rainfall estimates as a 100-minute accumulation starting between 90 and 30 min before the nominal time.
Here, we use the same approach to provide an evaluation and interpretation of the temporal characteristics for the IMERG estimates in terms of rain gauge accumulation on ground. The WEGN 5-min gridded data are used and integrated over accumulation times from 5 min (native sampling) to 100 min (twenty 5-min samples) for time offsets from $-20$ min to $+60$ min (in 5-min steps). Figure 9 shows the resulting RMSE of IMERG rainfall estimates versus WEGN data as a function of the gauge accumulation time, $\Delta$, and the time offset, $\delta$.
The minimum RMSE value for the IMERG-F estimates of the entire dataset (Fig. 9, top left) occurs at a $\Delta$ of about 25 min and a $\delta$ of about $+40$ min. This offset of 40 min exceeds the 30-min time resolution of IMERG, which means that the IMERG-F estimates are, on average, displaced by more than one time step. This suggests, for example, IMERG-F rainfall estimates during 09:00-09:30 UTC can be considered as gauge measurements during 09:40-10:05 UTC. The positive offset is consistent with the early bias in rainfall onset found in Sect. 4.2. The hot season shows a shorter offset for the minimum RMSE
compared to the warm season (Fig. 9, top middle and right), which agrees with the results of Villarini and Krajewski (2007).

Intercomparing the IMERG products for their common period in 2015 (Fig. 9, bottom row), it is visible that longer $\Delta$ values are needed to minimize RMSE of IMERG-E and IMERG-L rainfall estimates, while optimal $\delta$ values are obtained as around $+20$ min for the both datasets. In general, the IMERG NRT estimates show higher RMSE values compared to the IMERG-F estimates, as expected. Also they show relatively indistinct patterns and even multi-minimum RMSE values (in the case of
IMERG-E). As such, this approach of interpreting the rainfall estimates may not be sufficiently constrained by the NRT estimates, due to the limited sample size from only seven months of data and also due to larger errors. More years of data are needed before such an approach can provide a robust interpretation of the NRT estimates.

### 5  Conclusions

In this study, we evaluated half-hourly rainfall estimates from the IMERG-E, IMERG-L, and IMERG-F satellite data products
using gauge measurement data from the WEGN network in southeast Austria for the period of April–October in 2014 and 2015. The dense WEGN gauge network provided a unique opportunity for a direct grid-to-grid comparison over two selected IMERG 0.1° x 0.1° grid boxes. This evaluation work provides valuable insights and input to improve satellite rainfall retrieval



processes, to further inter-compare data among satellite-based rainfall products, and to achieve a better product quality in particular of NRT satellite rainfall data for different applications such as flood or landslides warning and agricultural forecasting or drought monitoring.

First, thorough statistical comparisons analyzing differences between IMERG estimates and WEGN data showed that the
IMERG-F run considerably outperforms the two NRT runs. IMERG-E and IMERG-L runs overestimate the rain rates at low intensities, leading to large discrepancies in accumulated rain volume, which result in a lower correlation with WEGN data in general. All three IMERG products tend to underestimates the rain rates at high intensities.

Second, the study of example rainfall events with distinct IMERG and WEGN discrepancies reveals specific situations, e.g., lack of PMW-based observations during short-term rainfall, when the IMERG runs can fail to describe rainfall features even
qualitatively. Here, again, we find significantly smaller errors in the IMERG-F estimates, by the monthly-gauge correction, compared to the IMERG NRT estimates.

Furthermore, by calculating the RMSE of the half-hourly IMERG satellite estimates against the WEGN ground based rainfall data as a function of gauge accumulation time and time offset, the minimum RMSE found for IMERG-F estimates suggests these can be regarded as a 25 min accumulation with a +40 min time offset (preceding the time of the gauge data by this time
span). Again, the results for the IMERG NRT estimates suggest significantly lower confidence, both due to insufficient sample size and larger estimation errors.

Consequently, our analysis across the different runs of IMERG demonstrates the effects of the additional processes on the final rainfall estimates. While the better performance of IMERG-F run is often attributed to the gauge adjustment procedure (Boushaki et al., 2009; Vila et al., 2009; Almazroui, 2011), we also identify the advantages of a greater number of
PMW-based estimates. On the other hand, the inclusion of forward and backward morphing in the IMERG-L run, with sparse PMW observations, provides only marginal benefits over the forward-only morphing in the IMERG-E run. In fact, our case study of example rainfall events illustrates the interesting possibility of cancellation in the backward and forward morphing estimates for the IMERG-L run, resulting in a performance poorer than in the IMERG-E run.

Further studies on detailed links between the errors in the final rainfall estimates and the upstream data sources or retrieval
processes, to alleviate those issues will contribute to improvements in the performance of the IMERG-L run (e.g., by accounting for time-lagged peaks or improving the cloud motion vectors) and, consequently, the IMERG-F run. Meanwhile, addressing instantaneous satellite estimates involved in the IMERG runs will help us to understand overestimation in the PMW estimates themselves.

Our future work on the evaluation of IMERG products will place one emphasis on the IMERG-F data, in order to better
understand the behavior of rainfall estimates with various conditions, such as different temporal accumulation, threshold, or involved PMW/IR sources. Using the WEGN high-resolution data, we can also explore rainfall uncertainty and variability at a IMERG subpixel-scale, another intriguing prospect. Additionally, it will be worthwhile to inter-compare the current version of IMERG-F data (V03) with the next version (V04 and V05), planned to be released later in 2017, to evaluate the improvements in the IMERG system.





*Acknowledgements.* The study was funded by the Austrian Science Fund (FWF) under research grant W 1256-G15 (Doctoral Programme Climate Change Uncertainties, Thresholds and Coping Strategies). Petersen and Tan acknowledge the NASA GPM/PMM Programs, and Tan acknowledges funding under the NASA Postdoctoral Program. The IMERG data are provided by the NASA/Goddard Space Flight Center's PMM and PSS teams through http://pmm.nasa.gov/data-access/downloads/gpm. WegenerNet funding is provided by the Austrian Ministry for Science and Research, the University of Graz, the state of Styria (which also included European Union regional development funds), and city of Graz; detailed information is found at www.wegcenter.at/wegenernet and the data are available through the WegenerNet data portal (www.wegenernet.org).



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




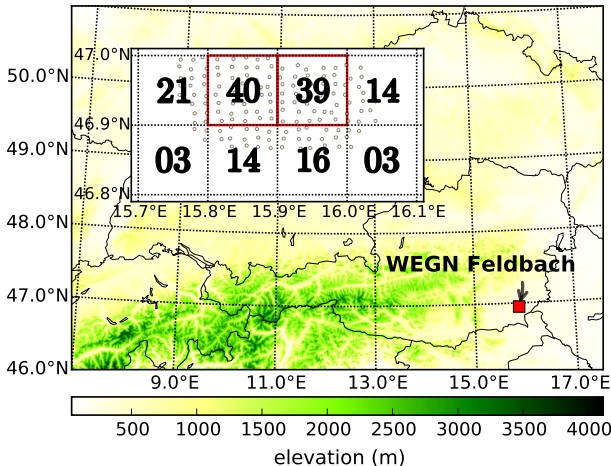

**Figure 1.** WEGN climate station network in the Feldbach region, southeast Austria. The inset plot shows an enlarged view of the number of WEGN rain gauges that are located within GPM IMERG 0.1° x 0.1° grid cells. The two red-framed grid boxes (within 46.9-47.0° N, 15.8-16.0° E), covered with 40 and 39 WEGN gauges each, are selected for the study, for brevity also termed "Grid 15.85" and "Grid 15.95" based on their mean longitude.

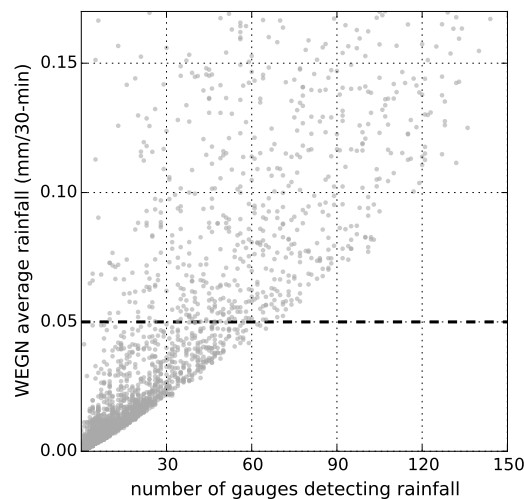

**Figure 2.** WEGN network-averaged half-hourly rainfall amounts as function of the number of WEGN gauges detecting rain ($\geq 0.1$ mm at single station). The rain amounts for half-hourly time intervals over the study period are shown, to check the reasonability of a rain/no-rain threshold for the IMERG data evaluation: 0.05 mm is used for the study.


**Table 1.** Description of the data: mean, standard deviation, maximum, number of rainfall amount values ($\geq 0.05$ mm 30-min$^{-1}$), and percentage of no-rain data ($< 0.05$ mm 30-min$^{-1}$). All half-hourly rainfall amounts for the period of April to October in 2014 and 2015 are used. IMERG-L and IMERG-E data are available from April 2015, i.e., used for the second year only.

|         | Mean (mm) | Standard Deviation (mm) | Max (mm) | Number of Rain Data | Percentage of No-rain (%) |
|---------|-----------|-------------------------|----------|---------------------|---------------------------|
| WEGN    | 0.82      | 1.29                    | 15.26    | 3,353               | 91.8                      |
| IMERG-F | 1.23      | 2.02                    | 25.61    | 2,671               | 93.5                      |
| IMERG-L | 1.33      | 2.82                    | 23.73    | 1,335               | 93.5                      |
| IMERG-E | 1.36      | 2.89                    | 25.67    | 1,248               | 93.9                      |

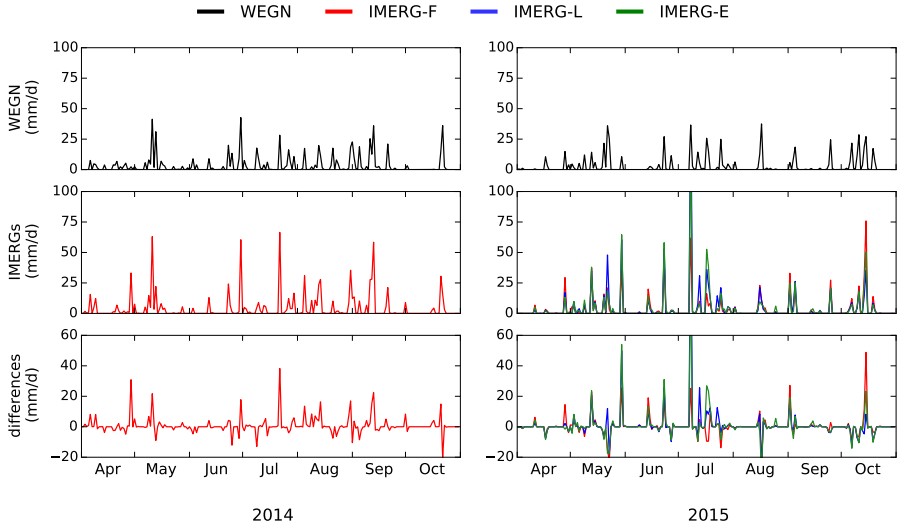

**Figure 3.** Accumulated 24-h rainfall (mm/d) over the selected study periods in 2014 (left) and 2015 (right), obtained from WEGN data (top) and IMERG estimates (middle), and differences of the IMERG estimates against WEGN data (bottom). Regarding IMERG, in the 2014 period only IMERG-F data are available.




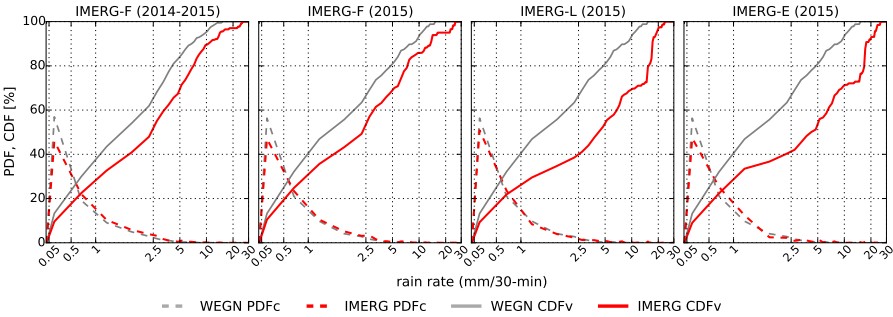

**Figure 4.** Occurrence probability density function of rain rates ($PDF_c$, dashed) and cumulative distribution functions of rain volume ($CDF_v$, solid) for WEGN (gray) and IMERG (red). Comparisons for IMERG-F for the full 2014-2015 period (left) and separately for IMERG-F (middle-left), IMERG-L (middle-right), and IMERG-E (right) for the 2015 period are shown.

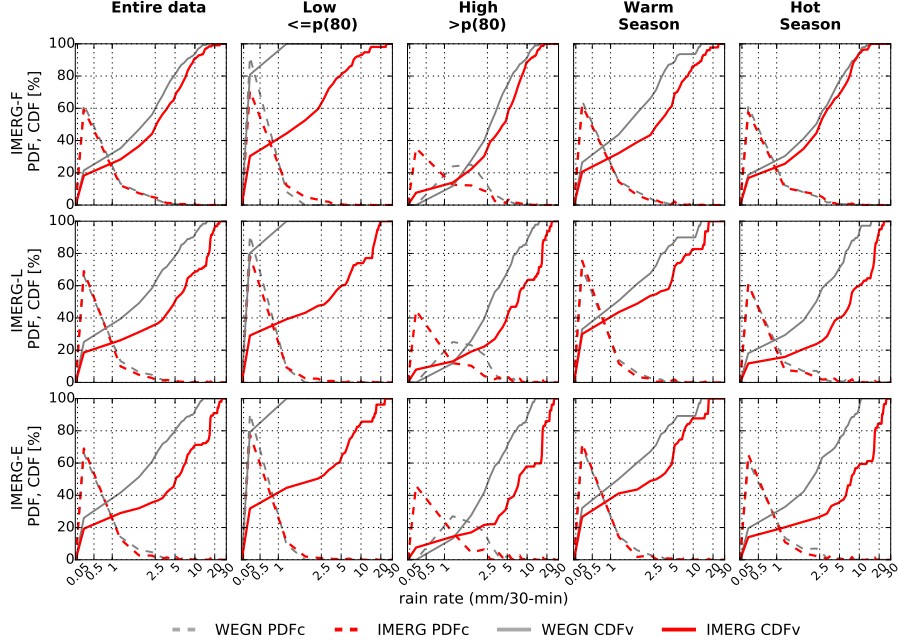

**Figure 5.** Same layout per panel as Fig. 4, but restricted to the data pairs for which IMERG and WEGN both detected rainfall ($\geq$ 0.05 mm 30-min[-1]). Comparisons are shown for IMERG-F (top), IMERG-L (middle), and IMERG-E (bottom), with using the entire data (left), the data divided into low and high rain amounts (middle-left, middle) based on the 80[th] percentile of WEGN data, or the data divided into two different seasons (middle-right, right), warm (April, May, October) and hot (June to September).





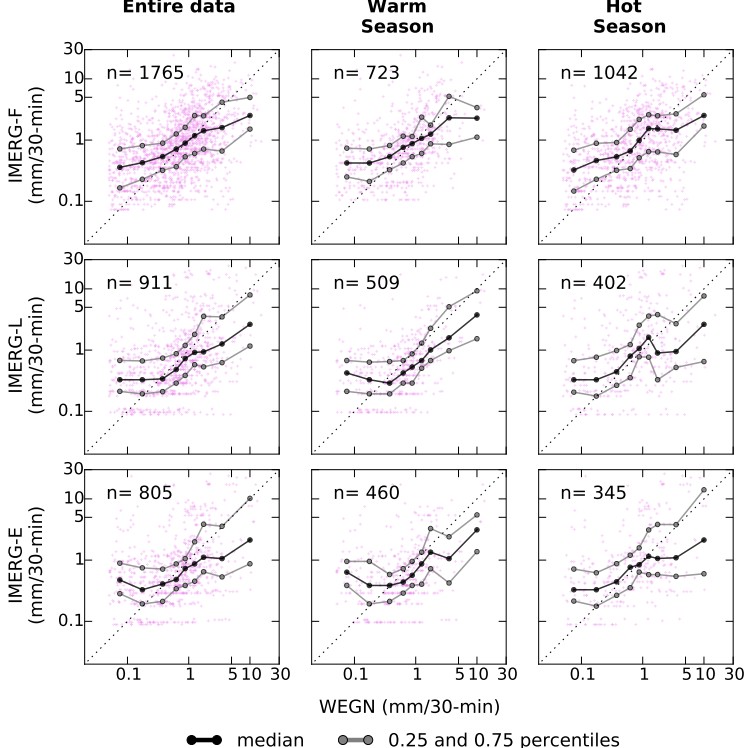

**Figure 6.** Scatter plots of half-hourly rainfall amounts for IMERG-F (top), IMERG-L (middle), and IMERG-E (bottom) versus WEGN, for the entire dataset (left), warm season only (middle), and hot season only (right). Overplotted are, for nine bins across the range with at least 30 IMERG-WEGN data pairs in each bin, the 50$^{th}$ percentile (black symbols and line) and the 25$^{th}$ and 75$^{th}$ percentiles (grey symbols and lines) of the IMERG estimates. Total number of data pairs (n) is indicated in the upper left of each panel. Note that a log-log scale is used.

**Table 2.** Validation statistics comparing the performance of the three IMERG datasets for the two selected grid cells covered by the WEGN.

|  |  | RB | MAE (mm) | NMAE | RMSE (mm) | NRMSE | r | $\rho$ |
|---|---|---|---|---|---|---|---|---|
| IMERG-F | Grid 15.85 | 0.22 | 1.18 | 0.97 | 2.19 | 1.79 | 0.34 | 0.46 |
|  | Grid 15.95 | 0.30 | 1.16 | 0.99 | 2.23 | 1.91 | 0.31 | 0.47 |
| IMERG-L | Grid 15.85 | 0.33 | 1.28 | 1.12 | 2.74 | 2.38 | 0.40 | 0.42 |
|  | Grid 15.95 | 0.54 | 1.41 | 1.29 | 3.31 | 3.02 | 0.30 | 0.44 |
| IMERG-E | Grid 15.85 | 0.28 | 1.27 | 1.11 | 2.84 | 2.47 | 0.38 | 0.38 |
|  | Grid 15.95 | 0.67 | 1.48 | 1.42 | 3.31 | 3.16 | 0.34 | 0.38 |





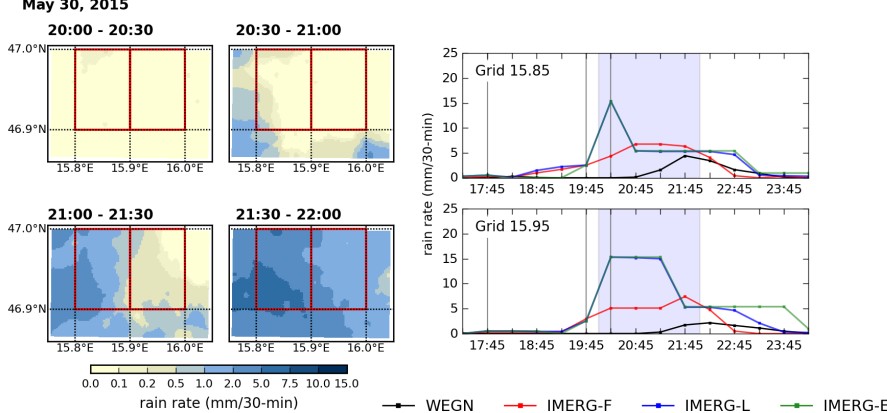

**Figure 7.** Example rainfall event in the warm season that occurred on 30 May 2015. (left) Spatial rainfall pattern over the WEGN network area for a sequence of four half-hourly WEGN rain rate maps, with the two selected IMERG grid cells indicated (red-framed boxes). (right) Time series of IMERG-F (red), IMERG-L (blue), IMERG-E (green) estimates and WEGN (black) data for the two IMERG grid cells over the period of the rainfall event, with the shaded areas highlighting the two hours illustrated by the map sequence to the left. Solid vertial lines indicate time steps where all three IMERG runs received PMW-based information for the rainfall retrieval at that time step, dotted vertical lines (not applicable for this event but for the event of Fig. 8 below) mean that either IMERG-E or both IMERG NRT runs did not receive PMW-based information at the time step, and no vertical line (the case for most time steps) implies that none of the IMERG runs received PMW-based information.

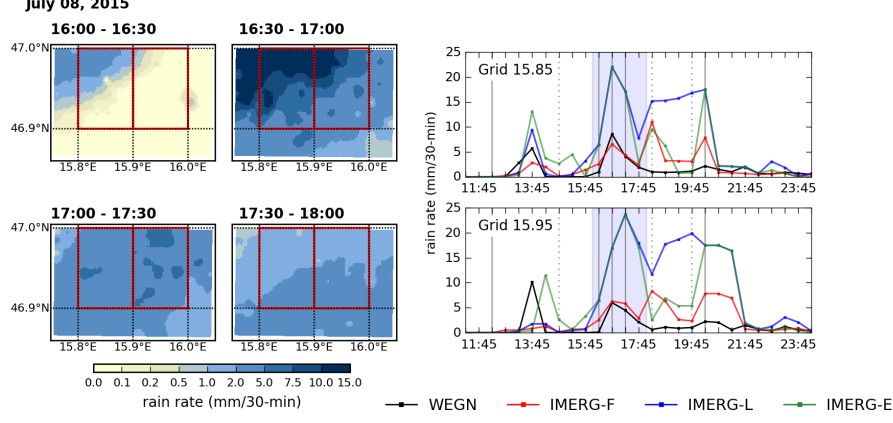

**Figure 8.** Same as Fig. 7, but for an example rainfall event in the hot season that occurred on 08 July 2015.




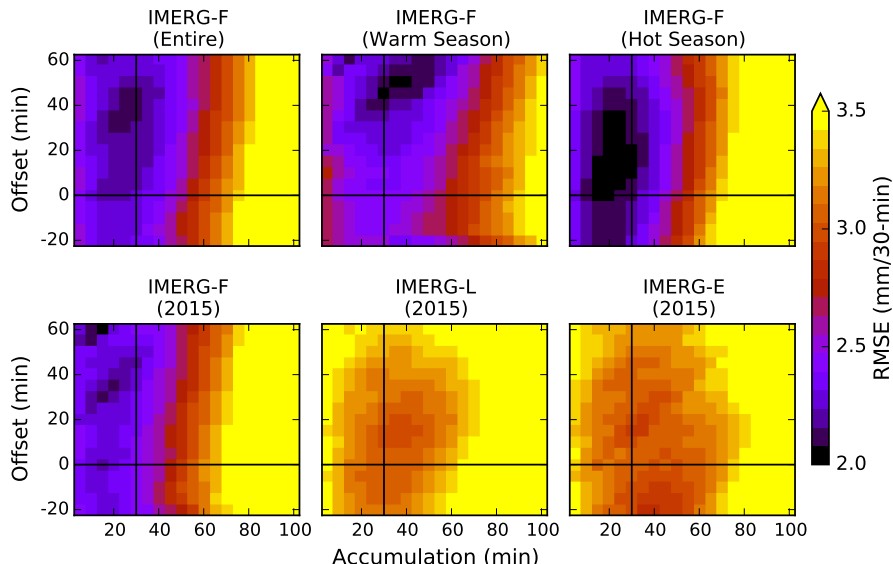

**Figure 9.** Contour plots of RMSE ($\mathrm{mm\,30\text{-}min^{-1}}$) between IMERG estimates and WEGN data as a function of WEGN gauge rainfall accumulation time and time offset (which means the time from which to start the WEGN gauge rainfall accumulation relative to the IMERG data start time).