# Peer review of "Evaluation of GPM IMERG Early, Late, and Final rainfall estimates with WegenerNet gauge data in southeast Austria"

_Hydrology and Earth System Sciences, 2017_

## Referee Comment (RC1) · Anonymous Referee #1 · 15 Jun 2017

Evaluation of GPM IMERG Early, Late, and Final rainfall estimates with WegenerNet gauge data in southeast Austria by Sungmin O et al.

The comparison study provides very useful information regarding the performance and issues of GPM IMERG products at the pixel level. Such high-density gauge network is very rare around the world for ground validation (GV) and the results should be very interesting not only for algorithm developers but also for other researchers and application users. The manuscript is well written and the research was well planned. However, I do have some concerns and questions that could improve the manuscript. For such reason, I recommend some major revisions are needed.

[Figure]

Major issues:

Fig. 2 is very useful to characterize the uniformity of gauge rainfall measurement. Is that possible to expand the rain rate above the current ∼0.16 mm/30-min?

In Figs. 7 and 8, is it possible to add the analysis for PMW-based and IR-based rainfall estimates by using HQprecipitation and IRprecipitation data that come with the three datasets (E, L, and Final Runs) and quantify the contribution from the two (PMW and IR) measurements? The co-author, J. Tan, has done such work before. So far, the conclusion was only based on the work done by others.

What can we do to correct the large positive biases in E and L Runs datasets since gauge data are not available at the time when the data are available?

The discussion part on some issues is missing. For example, are the results unique to Austria or can be applied to other places? It would be nice to discuss and compare your results with other studies that are already published. Major differences between v03 and v04 need to be discussed since v03 is obsolete. Morphing algorithm refinement, etc.

Minor:

Abstract: Please add the version number of the IMERG products to avoid confusion when new version is released in future. I assume it is Version 03.

P2. Line 5. Would be nice to add a description about the differences between the 3 IMERG datasets (E, L, and F runs) from the algorithm point of view in case readers are not familiar with these datasets.

Figure 1. What is the red dot in the map? Does it contain the two grids in red?

P3. Line 21. landslides => landslide?

P4. Line 1. Why? Any comments on the no major change?

Figure 2. What time period? It is not clear.

Table 1. Why don't have two seasons (warm and hot) for comparison?

P4. Line 27 the units are needed for 0.05

P. 6. Lines 12-17. Would be nice to give some weather conditions for both warm and hot seasons such as average surface air temperatures.

P. 8. Line 80. Any analysis on PMW observations? All datasets contain such parameters. I don't see any analysis here.

P. 9. Line 12. Why the time offset begins from -20 min not -60 min?

---

## Author Comment (AC1) · 22 Jun 2017

*We appreciate the comments and suggestions from Reviewer #1 which helped to improve the quality of this manuscript. We have addressed all issues indicated in the review report and provide a point-to-point response to the comments.*

**Major issues:**

**1 Fig. 2 is very useful to characterize the uniformity of gauge rainfall measurement. Is that possible to expand the rain rate above the current $\sim 0.16\ mm/30-min$?**
*We limited the y-axis to emphasize where our interest lies, i.e., very low rain rates. An inset plot will be added in Fig.2 (attached) to show higher rain rates.*

**2 In Figs. 7 and 8, is it possible to add the analysis for PMW-based and IR-based rainfall estimates by using HQprecipitation and IRprecipitation data that come with the three datasets (E, L, and Final Runs) and quantify the contribution from the two (PMW and IR) measurements? The co-author, J. Tan, has done such work before. So far, the conclusion was only based on the work done by others.**

*It is true that such data (HQ/IRprecipitation) can be used to analyze the contribution of each sensor to IMERG bias. However, given that we are more interested in comparing performance between three different IMERG runs, we approached the data in a slightly different way.*

*We examined how many HQprecipitation estimates were ingested into each IMERG run during the selected rain event, as indicated by vertical lines in Figures 7 and 8. For example, what we found in Fig.8 from HQprecipitation and IRprecipitation was that i) although IMERG Late uses more PMW estimates compared to IMERG Early, there is no difference in values of PMW- or IR-based estimates between IMERG Late and Early, and ii) IMERG Final shows a more significant difference (improvement) in its estimates after morphing or bias correction (i.e., precipitationCal/precipitationUncal), rather than in HQ/IRprecipitation.*

*Therefore, we analyzed data differences between IMERG runs from a view of applied algorithms. More explanation will be added on i) P.8, Line 32; "... all of which somehow overestimated the rain rates (no difference in data values between IMERG-E and IMERG-L once the data are collected from the same PMW sensor)...", and ii) P.10 Line 24; "... and the upstream data sources (i.e., contribution of each PMW/IR sensor to biases in IMERG estimates) ...".*

[Figure]

**3 What can we do to correct the large positive biases in E and L Runs datasets since gauge data are not available at the time when the data are available?**

*We appreciate the question. It seems to us that IMERG NRT errors will likely have to be dealt with at the algorithm-level (e.g. calculating accurate PMW or IR estimates), rather than at the user-level (e.g. applying bias correction) - unless the user has access to gauge data and can apply local corrections using data relevant to the NRT product. Therefore, at the moment, our results focus on i) providing accuracy information of IMERG NRT to data users, and ii) providing a benchmark point to see improvement in the next IMERG version.*

**4 The discussion part on some issues is missing. For example, are the results unique to Austria or can be applied to other places? It would be nice to discuss and compare your results with other studies that are already published.**

*The reviewer asks a good question. It does seem that the results should be transferable to other WEGN-like regions, i.e., land areas with moderate elevations. However, any direct comparison with other studies is limited because, to the authors' knowledge, there has been no research which evaluates all IMERG three runs; the independence of WEGN gauges (e.g., gauge data are not part of any gauge-adjusted IMERG dataset) enables us to conduct the evaluation uniformly across the IMERG runs.*

*This point will be addressed in Conclusion, P.10 after Line 23; ". . . These results on the performance of IMERG runs could be representative of other regions under similar conditions (i.e., mid-latitude land areas). The study approach is, however, not easily applicable to different precipitation regimes. This is mainly due to the limited availability of independent ground reference data like WEGN. As a result, WEGN offers valuable information about the accuracy of*

*IMERG estimates across its three different runs.".*

**5 Major differences between v03 and v04 need to be discussed since v03 is obsolete. Morphing algorithm refinement, etc.**

*The biggest difference in the new version would be the use of GPM intercalibrated data, rather than TRMM based intercalibrations. "IMERG V04 is the first version to use the GPM Core Observatory as a calibrator for the constellation satellite partners so it is expected to provide more consistent quality among the PMW/IR estimates." will be added at the end of Conclusion.*

**Minor:**

**1 Abstract: Please add the version number of the IMERG products to avoid confusion when new version is released in future. I assume it is Version 03.**

*"In this study, IMERG version 3 Early, Late, and Final . . ." will be in Abstract.*

**2 P2. Line 5. Would be nice to add a description about the differences between the 3 IMERG datasets (E, L, and F runs) from the algorithm point of view in case readers are not familiar with these datasets.**

*We believe that "2.1 GPM IMERG satellite rainfall estimates (P3, Lines 20 to 30)" can provide information on the differences between IMERG runs including applied algorithms (e.g., morphing scheme, gauge correction). We will add "See Section 2.1" when 'IMERG runs' first comes out in the manuscript - Introduction*

***P.2 Line 12.***

**3 Figure 1. What is the red dot in the map? Does it contain the two grids in red?**

*It was referring to the location of WegenerNet network; Feldbach in Austria. We assume the red color made it confusing, so we have converted the mark in black (new figure is attached). Also the caption of Fig.01 will be; "... the Feldbach region (black square) ...".*

**4 Line 21. landslides => landslide?**

*Thank you for the correction.*

**5 P4. Line 1. Why? Any comments on the no major change?**

*First, as we mentioned, preliminary intercomparison between IMERG V03 and V04 did not identify any major changes in performance, so it is unlikely that our conclusions will change. Second, most of the changes in V04 (see V04 IMERG Final Run Release Notes; https://pps.gsfc.nasa.gov/Documents/IMERG_FinalRun_V04_release_notes.pdf) are applied to all three IMERG runs, so any improvements to the Final run should also involve a similar improvement to the Early and Late runs. Furthermore, it is likely that the algorithmic and data differences between the runs (e.g. use of backward morphing between Early and Late runs, the use of gauges in the Final runs) have a stronger influence than any of these changes.*

*This explanation will be added at the end of "2.1 GPM IMERG satellite rainfall*

*estimates" (P.4)*

**6 Figure 2. What time period? It is not clear.**

*The time period is same as the study period. "... (April to October in 2014-2015)..." will be added in the caption of Figure 2.*

**7 Table 1. Why don't have two seasons (warm and hot) for comparison?**

*This table is intended to provide general information on data sets used in the study, rather than comparison results. In addition, some information (e.g., max value, number of rain data for each season) can be obtained from the Fig. 5 and Fig. 6. Therefore, we would prefer the table as it is for the readability of the manuscript.*

**8 P4. Line 27 the units are needed for 0.05**

*'$\mathrm{mm}/30 - \mathrm{min}^{-1}$' will be added.*

**9 P. 6. Lines 12-17. Would be nice to give some weather conditions for both warm and hot seasons such as average surface air temperatures.**

*Thank you for the suggestion. "According to temperature measurements collected by WEGN, the average 2-m air temperature of the study period (2014-2015) was 12.2 °C in the warm season and 18.6 °C in the hot season" will be added at P.6 Lines 15.*

**10 P. 8. Line 80. Any analysis on PMW observations? All datasets contain such parameters. I don't see any analysis here.**

*The focus of this particular study was on the number of PMW observations used in each IMERG run during the events (e.g., P.8 Line 15, Lines 31-33). It is beyond our scope to examine the performance of the different PMW observations contributing to the IMERG estimates. Please also refer to the reply #2 in Main issue.*

**11 Line 12. Why the time offset begins from -20 min not -60 min?**

*This is because the best agreement (lower RMSE) between data is found in a positive Offset region (i.e., above 0-min), so this time range is selected to 'zoom in' at where the minima occur.*

**Fig. 1.**

Fig. 2.

[Figure]

---

## Referee Comment (RC2) · Anonymous Referee #2 · 1 Jul 2017

This study evaluates the GPM IMERG products using rain gauge data from a dense network in Austria. The three IMERG products (Early, Late, and Final) are compared with the areal average of rain gauge data over two IMERG grid cells of 0.1° × 0.1°. The authors perform statistical analyses to quantitatively and qualitatively define errors in the IMERG products and also visually inspect two example rainfall events for diagnosing some detailed performance of the IMERG estimates. They conclude that "the IMERG-Final estimates are in the best agreement with the WEGN data, particularly for the hot season."

I think that this is an interesting paper dealing with an early evaluation of the IMERG

products, and the topic of GPM Ground Validation (GV) is suitable for Hydrology and Earth System Sciences. I also think that this study provides useful information and some insight for algorithm developers as well as hydrologic users. However, I have some major concerns and questions, and the manuscript needs to provide some more clear insight and discussions on the findings. I would recommend this manuscript for publication after some moderate revisions. My detailed comments are provided below:

Major comments:

1. Gauge representativeness I think that 40 and 39 for given $0.1° \times 0.1°$ (roughly 10 $\times$ 10 km2) grid cells are definitely good numbers of gauges for estimating areal average rainfall at a typical temporal scale of satellite rainfall products (e.g., three-hourly or monthly). However, since this study evaluates 30-min products for which random variability is much higher, I think that the authors should justify that the gauge representativeness error is not significant at the space and time scales used in this study. The authors could provide the structure of spatial correlation and variance reduction for the study area as shown in Villarini and Krajewski (2007).

I also think that the use of 200 $\times$ 200 m2 gridded rainfall data at 5-min scale is not reasonable (but this does not significantly affect the results of this study because the gridded data are aggregated over 30-min and 10 $\times$ 10 km2 scales and then used). The 5-min gridded map contains so much variability (in terms of gauge representativeness) due to high space and time scales used as well as the tipping bucket rain gauge error itself with tip counts within 5-min (this will also decrease with longer time integration).

2. Gauge data independence Please clarify that the rain gauge data used in this study are from an independent network. The authors state that "the WEGN is not a member network of the GPCC network" (Page 4 Line 32). However, there is another statement (Page 7 Line 33 – Page 8 Line 2) that a better performance may be attributed to an Austrian national station that are associated with the GPCC product. These are confusing.

3. Figure 5 (the second and third columns) I may miss something, but the WEGN PDFs and CDFs exist for the entire rain rate regions although they are presented for low (<1.2 mm) and high (>1.2mm) rain amounts each. For example, shouldn't the PDFs and CDFs start from 1.2 mm in the third column (high rain rate)? Why do the PDFs and CDFs exist for R<1.2 mm?

4. Time shift in Figures 7 and 8 It is hard to say that the observed patterns in Figures 7 and 8 show a time shift. I think that we can say there is a time shift only when the rainfall durations are the same between reference (WEGN) and IMERG products and starting times are different. It seems to me like that the observed patterns are just errors, probably by morphing and other reasons. In Figure 8, the shapes and peak times are all different and it is hard to find any consistent or systematic tendency.

Minor comments:

1. Page 3 Line 1. "…. has been suggested to guarantee a monthly error of under 10%". This cannot be directly applied to this study because of the temporal scale difference (monthly vs. 30-min). The gauge representativeness is a function of space and time scales used (Seo and Krajewski 2010).

Reference: Seo, B.-C., and W. F. Krajewski (2010), Scale dependence of radar-rainfall uncertainty: Initial evaluation of NEXRAD's new super-resolution data for hydrologic applications, Journal of Hydrometeorology, 11(5), 1191-1198.

2. Page 4 Line 24. Please explain "high native time resolution."

3. Page 4 Line 27. Please clarify the rainfall threshold used in this study. In the caption of Figure 2, there is a phrase "$\geq$ 0.1 mm at single station". How big is the tip resolution against the threshold?

4. Page 6 Line 33. Please clarify the difference between Figures 4 and 5. Is the threshold 0.05 mm applied to both Figures 4 and 5?

5. Page 7 Line 5. Isn't it 0.05 instead of 0.5?

6. Page 7. Line 6. Please explain "the entire WEGN data." Is it without applying the threshold?

7. Page 7 Line 31. It would be useful if the correlation coefficient values obtained from other evaluation studies are provided and compared with those of this study.

8. Page 9 Line 2-4. What (more PMW estimates or rain gauge correction) has more contribution to the improvement? I think that this is very important point in the satellite product evaluation.

9. Section 4.3. Please add some implication of the result found in Section 4.3 and discussions on how to use the products for hydrologic applications.

---

## Referee Comment (RC3) · R. Uijlenhoet (Referee) · 14 Jul 2017

This manuscript provides a detailed evaluation of the three available IMERG products using a high-resolution rain gauge network in Austria. These products have recently become available to the (hydrological) user community, following the launch of the GPM core satellite. Careful evaluations such as the one presented in this manuscript are of great value to end-users wanting to employ these products in (hydrological) applications. In that sense, the manuscript is both timely and topical and I recommend its publication in HESS following minor revisions.

Basically, I have little to add to the review reports submitted by the other referees, to

which the authors have responded in a satisfactory manner, as far as I am concerned. The authors are apparently unaware of a recent evaluation of one of the IMERG products over The Netherlands, see "First-Year Evaluation of GPM Rainfall over the Netherlands: IMERG Day 1 Final Run (V03D)" by Rios Gaona et al., published in Journal of Hydrometeorology (2016). Perhaps that paper contains relevant material for the authors to compare their results to. This is particularly relevant if the authors want to discuss the applicability of their results to other regions, an issue referred to also by one of the other reviewers.

Finally, I look forward to seeing an actual hydrological evaluation of the IMERG products considered by the authors, e.g. used as forcing in rainfall-runoff models of one or more river catchments in Austria.

Remko Uijlenhoet
* * *

---

## Author Comment (AC2) · 14 Jul 2017

*We appreciate your comments and suggestions, which helped to enhance the quality of this manuscript. We have discussed all issues indicated in the review report and provide a point-to-point response to the comments below.*

**Major comments:**

**1 Gauge representativeness I think that 40 and 39 for given $0.1° × 0.1°$ (roughly $10 × 10$ km2) grid cells are definitely good numbers of gauges for estimating areal average rainfall at a typical temporal scale of satellite rainfall products (e.g., three-hourly or monthly). However, since this study evaluates 30-min products for which random variability is much higher, I think that the authors should justify that the gauge representativeness error is not significant at the space and time scales used in this study. The authors could provide the structure of spatial correlation and variance reduction for the study area as shown in Villarini and Krajewski (2007). I also think that the use of $200 × 200$ m2 gridded rainfall data at 5-min scale is not reasonable (but this does not significantly affect the results of this study because the gridded data are aggregated over 30-min and $10 × 10$ km2 scales and then used). The 5-min gridded map contains so much variability (in terms of gauge representativeness) due to high space and time scales used as well as the tipping bucket rain gauge error itself with tip counts within 5-min (this will also decrease with longer time integration).**

> *=> We thank the reviewer for this comment and address as follows: Given the spatially uniform and high resolution configuration of WEGN, and additionally considering that the network region is characterized by low relief, we can ensure that WEGN provides a reliable ground reference, particularly in terms of spatial representativeness for IMERG validation.*
>
> *Fig.1 below plots VRF (variance reduction factor) of WEGN 30-min data at Grid 15.85 (the pixel with 40 gauges) as a function of number of gauges randomly sub-sampled from the 40 gauges. Based on this result, we will add the following sentence in the : "The Variance Reduction Factor (VRF, Villarini et al. 2008), examined with half-hourly WEGN data of the 40-gauges grid box, is about 0.02 for 10 gauges (average of 40 random combinations) and little or no improvement is observed in the VRF beyond 10 gauges." (Page 3, Lines 1)*

[Figure]

**Fig. 1 VRF for different gauge densities, calculated with 40 random combinations for each gauge number. Filled circles represent the average of VRF for a combination. Vertical bars show the associated min-max range.**

*Fig.2 below shows normalized RMSE of WEGN mean areal rainfall (the average of all 40 gauges is assumed to be true) as a function of gauge number for 30-min and 5-min accumulation times, for Grid 15.85. The errors are calculated at a time step of maxima (left panels) and median (right panels) rainfall spatial standard deviation (when more than 20 gauges detected rainfall). Each boxplot is based on 3,000 possible combinations (with the exception of gauge number 1 and 39, based on 40 possible combinations; and gauge number 2 and 38, based on 780 possible combinations).*

[Figure]

**Fig. 2 Dependence of uncertainty of IMERG pixel-averaged rainfall estimation on the number of gauges. Note that the ranges of y-axis vary between plots.**

*It is clearly shown that the spatial uncertainty of mean areal rainfall substantially decreases with increasing gauge numbers (little change from ~10 gauges, as expected from the VRF), and areal rainfall estimates tend to converge toward the 'true' value. As the reviewer pointed out, 5-min data contain higher rainfall variability but, nevertheless, the figure shows that we can still obtain reliable mean areal rainfall from WEGN for the study domain, with a significant decrease of spatial representation errors.*

**2 Gauge data independence Please clarify that the rain gauge data used in this study are from an independent network. The authors state that "the WEGN is not a member network of the GPCC network" (Page 4 Line 32). However, there is another statement (Page 7 Line 33 – Page 8 Line 2) that a better performance may be attributed to an Austrian national station that are associated with the GPCC product. These are confusing.**

> *=> We apologize for the confusion. There are two different networks in the study domain: 1) 40+39 WEGN stations and 2) an Austrian national weather station from the network of the national meteorological service (ZAMG). This ZAMG station is a member of the GPCC network. We did not include the station for our study since its data directly affect the performance of IMERG Final run.*
>
> *We will rewrite as "… attributed to an Austrian national station, which is not part of the WEGN gauges, but located within the WEGN area (in Grid cell 15.85), of which measurements are integrated into the GPCC …" (Page 8 Line 1), so that readers can understand it clearly.*

**3 Figure 5 (the second and third columns) I may miss something, but the WEGN PDFs and CDFs exist for the entire rain rate regions although they are presented for low (1.2mm) rain amounts each. For example, shouldn't the PDFs and CDFs start from 1.2 mm in the third column (high rain rate)? Why do the PDFs and CDFs exist for R<1.2mm?**

> *=> Your understanding is correct. WEGN PDFs and CDFs have values only from/till 1.2 mm, but current plots contain lines extending beyond stated range because, for example, WEGN PDFc in the high rain rate cases (third column) starts from 0% at 0.05 mm (not at 1.2 mm) so we get to see values even in < 1.2 mm ranges. We now re-plotted the second and third columns so the starting/end points of WEGN PDFs and CDFs will be indicated by a vertical line and the WENG PDFc/CDFv end/begin at these lines. Thank you for the question.*

[Figure]

 #4 Time shift in Figures 7 and 8 It is hard to say that the observed patterns in Figures 7 and 8 show a time shift. I think that we can say there is a time shift only when the rainfall durations are the same between reference (WEGN) and IMERG products and starting times are different. It seems to me like that the observed patterns are just errors, probably by morphing and other reasons. In Figure 8, the shapes and peak times are all different and it is hard to find any consistent or systematic tendency.

> => **Thank you for pointing this out. The term "shift" is used to refer to the time disagreement between WEGN and IMERG; to be clearer, we will re-write the sentence (Page 8 Line13-14) as "… with a time difference of about 2 hours earlier in starting time. This false alarm suggests that…"**

**Minor comments:**

**1 Page 3 Line 1. ". . .. has been suggested to guarantee a monthly error of under 10%". This cannot be directly applied to this study because of the temporal scale difference (monthly vs. 30-min). The gauge representativeness is a function of space and time scales used (Seo and Krajewski 2010).**

> => **We will add the VRF value obtained from WEGN 30-min data in the  (Page 3 Line 1) as you suggested, please also refer to Major comment #1.**

**2 Page 4 Line 24. Please explain "high native time resolution."**

> => **We will modify the sentence as "…, WEGN gridded data with 5-min native resolution are used." (Page 4 Line 24)**

**3 Page 4 Line 27. Please clarify the rainfall threshold used in this study. In the caption of Figure 2, there is a phrase "≥ 0.1 mm at single station". How big is the tip resolution against the threshold?**

> *=> The caption means that we applied 0.1 mm threshold to count the number of WEGN gauges (the tip resolution is 0.1 mm) that detected rainfall. So this threshold is used only in Figure 2. On the other hand, 0.05 mm threshold is the one used to define rain/no-rain data in each grid box (average of multiple gauges) for the study. We will modify the Figure 2 caption to say "(gauges with >= 0.1mm rainfall are counted)" to make it clearer.*

**4 Page 6 Line 33. Please clarify the difference between Figures 4 and 5. Is the threshold 0.05 mm applied to both Figures 4 and 5?**

> *=> Yes, the same threshold is applied. Figure 4 shows all rain data of IMERG and WEGN (i.e. not only hits but also false alarms and misses), whereas Figure 5 shows only matched data (i.e., only hits; this is explained at Page 6 Line 33 - Page7 Line 1).*

**5 Page 7 Line 5. Isn't it 0.05 instead of 0.5?**

> *=> It is 0.5. We are comparing the POD score of IMERG-WEGN for all rain intensity ranges versus higher intensities (for WEGN > 0.5) to show that misses of IMERG are observed at relatively low rain intensities. To make it clear, we will re-write the sentence of Page 7 Lines 4- 6: "Indeed, we recomputed the POD score using only values when WEGN is above 0.5 mm 30-$min^{-1}$ and found that the PODs are 0.70, 0.79, and 0.75 for IMERG-E, IMERG-L, and IMERG-F, …". Please also refer to the next comment – Minor comment #6.*

**6 Page 7. Line 6. Please explain "the entire WEGN data." Is it without applying the threshold?**

> *=> It is with applying the threshold. Here "the entire WEGN data" means all available WEGN data; we will correct it as "… against WEGN for the entire range of rain intensities above the 0.05 mm 30-$min^{-1}$ threshold" in order to make it clear.*

**7 Page 7 Line 31. It would be useful if the correlation coefficient values obtained from other evaluation studies are provided and compared with those of this study.**

> *=> It seems to us that it will be very difficult to directly compare our correlation coefficient values with those of other studies, because the agreement score between ground reference and IMERG can be very different depending on i) the nature of the ground reference (e.g., radar vs gauge, number of gauges, rainfall regime) and on ii) temporal and spatial scales used*

*for comparison. Moreover, given that the study aims to compare the performance between three different IMERG runs, we show the indices in Table 2 to explore differences between the IMERG runs rather than to give absolute scores of each run.*

**8 Page 9 Line 2-4. What (more PMW estimates or rain gauge correction) has more contribution to the improvement? I think that this is very important point in the satellite product evaluation**

> *=> Thank you for the suggestion. We will add "From these two case studies, it appears that the gauges provide a greater improvement to IMERG Final estimates" after Page 9 Line 4.*

**9 Section 4.3. Please add some implication of the result found in Section 4.3 and discussions on how to use the products for hydrologic applications.**

> *=> We are grateful for this comment. We will add "This analysis identifies possible sources of error that should be considered in the context of hydrological applications of IMERG data. For instance, biases (overestimation in this case) in IMERG rainfall estimates will inevitably propagate through hydrologic models, and consequently this would lead to greater errors in runoff. The magnitude of biases can be reduced when IMERG Final estimates are used. Time offset bias, however, remains relatively stable across all three IMERG runs, especially in the warm season. Therefore, comparison or adjustment of IMERG estimates using local ground reference (if available) in terms of biases not only in amount of rainfall but also in its timing should be considered as an approach to meet the required level of accuracy in rain data." in <Section 4.3> (page 9, Line 23)*

---

## Author Comment (AC3) · 17 Jul 2017

We appreciate the positive feedback from the reviewer. We will cite the paper (Rios Gaona et al., 2016) to provide an example of recent study on the GPM IMERG (P2, Line 15), and also to support our discussion on the effect of gauge adjustment (P9, Line 4). Thank you for suggesting the paper. Regarding the hydrological evaluation, some runoff modeling studies with IMERG have been done in the U.S., China, and actually other locations around the globe, but, we agree that this would be a worthy endeavor for Austrian catchments.
256, 2017.

---

## Author Response (AR1)

**Response to the comments of Referee #1 on* "Evaluation of GPM IMERG Early, Late, and Final rainfall estimates with WegenerNet gauge data in southeast Austria" *by* Sungmin O et al.**

*We would like to thank you for your careful review of our manuscript. We have already provided a point-by-point response to the comments during the Interactive Discussion (Link), but here the page and line numbers where the corresponding revision can be found in the Marked Manuscript are updated.*

**Major issues:**

**1 Fig. 2 is very useful to characterize the uniformity of gauge rainfall measurement. Is that possible to expand the rain rate above the current ~0.16 mm/30-min?**

> *=> We limited the y-axis to emphasize where our interest lies, i.e., very low rain rates. An inset plot is added in Fig. 2 (Page 16) to show higher rain rates.*

**2 In Figs. 7 and 8, is it possible to add the analysis for PMW-based and IR-based rainfall estimates by using HQprecipitation and IRprecipitation data that come with the three datasets (E, L, and Final Runs) and quantify the contribution from the two (PMW and IR) measurements? The co-author, J. Tan, has done such work before. So far, the conclusion was only based on the work done by others.**

> *=> It is true that such data (HQ/IRprecipitation) can be used to analyze the contribution of each sensor to IMERG bias. However, given that we are more interested in comparing performance between three different IMERG runs, we approached the data in a slightly different way.*
>
> *We examined how many HQprecipitation estimates were ingested into each IMERG run during the selected rain event, as indicated by vertical lines in Figures 7 and 8. For example, what we found in Fig.8 from HQprecipitation and IRprecipitation was that i) although IMERG Late uses more PMW estimates compared to IMERG Early, there is no difference in values of PMW- or IR-based estimates between IMERG Late and Early, and ii) IMERG Final shows a more significant difference (improvement) in its estimates after morphing or bias correction (i.e., precipitationCal/precipitationUncal), rather than in HQ/IRprecipitation.*
>
> *Therefore, we analyzed data differences between IMERG runs from a view of applied algorithms. More explanation is added on i) P.9, Lines 11-12; "(no difference in data values*

*between IMERG-E and IMERG-L once the data are collected from the same PMW sensor)", and*
*ii) P.11, Line 16; "(i.e., contribution of each PMW/IR sensor to biases in IMERG estimates)".*

**3 What can we do to correct the large positive biases in E and L Runs datasets since gauge data are not available at the time when the data are available?**

> *=> We appreciate the question. It seems to us that IMERG NRT errors will likely have to be dealt with at the algorithm-level (e.g. calculating accurate PMW or IR estimates), rather than at the user-level (e.g. applying bias correction) – unless the user has access to gauge data and can apply local corrections using data relevant to the NRT product. Therefore, at the moment, our results focus on i) providing accuracy information of IMERG NRT to data users, and ii) providing a benchmark point to see improvement in the next IMERG version.*

**4 The discussion part on some issues is missing. For example, are the results unique to Austria or can be applied to other places? It would be nice to discuss and compare your results with other studies that are already published.**

> *=> The reviewer asks a good question. It does seem that the results should be transferable to other WEGN-like regions, i.e., land areas with moderate elevations. However, any direct comparison with other studies is limited because, to the authors' knowledge, there has been no research which evaluates all IMERG three runs. The independence of WEGN gauges (i.e., gauge data are not part of any gauge-adjusted IMERG dataset) enables us to conduct the evaluation uniformly across the IMERG runs.*
>
> *This point is addressed in <Conclusion>, P.11, Lines 10-14; "These results on the performance of IMERG runs could be representative of other regions under similar conditions (i.e., mid-latitude land areas). The study approach is, however, not easily applicable to different precipitation regimes. This is mainly due to the limited availability of independent ground reference data like WEGN. As a result, WEGN offers valuable information about the accuracy of IMERG estimates across its three different runs."*

**#5** Major differences between v03 and v04 need to be discussed since v03 is obsolete. Morphing algorithm refinement, etc.

> *=> The biggest difference in the new version would be the use of GPM intercalibrated data, not TRMM based intercalibrations. The sentence of "IMERG V04 is the first version to use the GPM Core Observatory as a calibrator for the constellation satellite partners so it is expected to provide more consistent quality among the PMW/IR estimates." is added at the end of <Conclusion> (Page 11).*

**Minor:**

**1 Abstract: Please add the version number of the IMERG products to avoid confusion when new version is released in future. I assume it is Version 03.**

> *=> Added. Abstract Line 5.*

**2 P2. Line 5. Would be nice to add a description about the differences between the 3 IMERG datasets (E, L, and F runs) from the algorithm point of view in case readers are not familiar with these datasets.**

> *=> We believe that <2.1 GPM IMERG satellite rainfall estimates> (P3, Lines 24-33) can provide information on the differences between IMERG runs including applied algorithms (e.g., morphing scheme, gauge correction). We have added "(see Sect. 2.1 for details)" when 'IMERG runs' first comes out in the manuscript. Page 2, Line 12.*

**3 Figure 1. What is the red dot in the map? Does it contain the two grids in red?**

> *=> It was referring to the location of WegenerNet network; Feldbach in Austria. We assume the red color made it confusing, so we have converted the mark into black. Also "the Feldbach region (black square)" has been added in the caption of Fig.01 (Page 16).*

**4 P3. Line 21. landslides => landslide?**

> *=> Thank you for the correction.*

**5 P4. Line 1. Why? Any comments on the no major change?**

> *=> First, as we mentioned, preliminary intercomparison between IMERG V03 and V04 did not identify any major changes in performance, so it is unlikely that our conclusions will change. Second, most of the changes in V04 (see V04 IMERG Final Run Release Notes; https://pps.gsfc.nasa.gov/Documents/IMERG_FinalRun_V04_release_notes.pdf) are applied to all three IMERG runs, so any improvements to the Final run should also involve a similar improvement to the Early and Late runs. Furthermore, it is likely that the algorithmic and data differences between the runs (e.g. use of backward morphing between Early and Late runs, the use of gauges in the Final runs) have a stronger influence than any of these changes.*
>
> > *This explanation is added at the end of <2.1 GPM IMERG satellite rainfall estimates>; Page 4, Lines 5-9.*

**6 Figure 2. What time period? It is not clear.**

> *=> The time period is same as the study period. "(April to October in 2014-2015)" is added in the caption of Figure 2 (Page 16).*

**7 Table 1. Why don't have two seasons (warm and hot) for comparison?**

> *=> This table is intended to provide general information on data sets used in the study, rather than comparison results. In addition, some information (e.g., max value, number of rain data for each season) can be obtained from the Fig. 5 and Fig. 6. Therefore, we would prefer the table as it is for the readability of the manuscript.*

**8 P4. Line 27 the units are needed for 0.05**

> *=> mm/30-min$^{-1}$ is added (Page 4, Line 34).*

**9 P. 6. Lines 12-17. Would be nice to give some weather conditions for both warm and hot seasons such as average surface air temperatures.**

> *=> Thanks for the suggestion. We have added "According to temperature measurements collected by WEGN, the average 2-m air temperature of the study period (2014-2015) was 12.2 °C in the warm season and 18.6 °C in the hot season"; Page 6, Lines 24-25.*

**10 P. 8. Line 80. Any analysis on PMW observations? All datasets contain such parameters. I don't see any analysis here.**

> *=> The focus of this particular study was on the number of PMW observations used in each IMERG run during the events (e.g., P.8, Lines 27-29 and P.9, Lines 10-13). It is beyond our scope to examine the performance of the different PMW observations contributing to the IMERG estimates. Please also refer to the reply #2 in Main issue.*

**11 P. 9. Line 12. Why the time offset begins from -20 min not -60 min?**

> *=> This is because the best agreement (lower RMSE) between data is found in a positive Offset region (i.e., above 0-min), so this time range is selected to 'zoom in' at where the minima occur.*

**Response to the comments of Referee #2 on* "Evaluation of GPM IMERG Early, Late, and Final rainfall estimates with WegenerNet gauge data in southeast Austria" *by* Sungmin O et al.**

*We greatly appreciate your comments and suggestions. We have already provided a point-by-point response to the comments during the Interactive Discussion (Link), but here the page and line numbers where the corresponding revision can be found in the Marked Manuscript are updated.*

**Major comments:**

**1 Gauge representativeness I think that 40 and 39 for given 0.1◦ × 0.1◦ (roughly 10 × 10 km2) grid cells are definitely good numbers of gauges for estimating areal average rainfall at a typical temporal scale of satellite rainfall products (e.g., three-hourly or monthly). However, since this study evaluates 30-min products for which random variability is much higher, I think that the authors should justify that the gauge representativeness error is not significant at the space and time scales used in this study. The authors could provide the structure of spatial correlation and variance reduction for the study area as shown in Villarini and Krajewski (2007). I also think that the use of 200 × 200 m2 gridded rainfall data at 5-min scale is not reasonable (but this does not significantly affect the results of this study because the gridded data are aggregated over 30-min and 10 × 10 km2 scales and then used). The 5-min gridded map contains so much variability (in terms of gauge representativeness) due to high space and time scales used as well as the tipping bucket rain gauge error itself with tip counts within 5-min (this will also decrease with longer time integration).**

> => *We thank the reviewer for this comment and address as follows: Given the spatially uniform and high resolution configuration of WEGN, and additionally considering that the network region is characterized by low relief, we can ensure that WEGN provides a reliable ground reference, particularly in terms of spatial representativeness for IMERG validation.*
>
> *Fig.1 below plots VRF (variance reduction factor) of WEGN 30-min data at Grid 15.85 (the pixel with 40 gauges) as a function of number of gauges randomly sub-sampled from the 40 gauges. Based on this result, we have added the following sentence in the : "The Variance Reduction Factor (VRF) (Villarini et al. 2008), examined with half-hourly WEGN data of the 40-gauges grid box, is about 0.02 for 10 gauges (average of 40 random combinations) and little or no improvement is observed in the VRF beyond 10 gauges." (Page 3, Lines 3-5)*

[Figure]

**Fig. 1 VRF for different gauge densities, calculated with 40 random combinations for each gauge number. Filled circles represent the average of VRF for a combination. Vertical bars show the associated min-max range.**

*Fig.2 below shows normalized RMSE of WEGN mean areal rainfall (the average of all 40 gauges is assumed to be true) as a function of gauge number for 30-min and 5-min accumulation times, for Grid 15.85. The errors are calculated at a time step of maxima (left panels) and median (right panels) rainfall spatial standard deviation (when more than 20 gauges detected rainfall). Each boxplot is based on 3,000 possible combinations (with the exception of gauge number 1 and 39, based on 40 possible combinations; and gauge number 2 and 38, based on 780 possible combinations).*

[Figure]

**Fig. 2 Dependence of uncertainty of IMERG pixel-averaged rainfall estimation on the number of gauges. Note that the ranges of y-axis vary between plots.**

*It is clearly shown that the spatial uncertainty of mean areal rainfall substantially decreases with increasing gauge numbers (little change from ~10 gauges, as expected from the VRF), and areal rainfall estimates tend to converge toward the 'true' value. As the reviewer pointed out, 5-min data contain higher rainfall variability but, nevertheless, the figure shows that we can still obtain reliable mean areal rainfall from WEGN for the study domain, with a significant decrease of spatial representation errors.*

**2 Gauge data independence Please clarify that the rain gauge data used in this study are from an independent network. The authors state that "the WEGN is not a member network of the GPCC network" (Page 4 Line 32). However, there is another statement (Page 7 Line 33 – Page 8 Line 2) that a better performance may be attributed to an Austrian national station that are associated with the GPCC product. These are confusing.**

*=> We apologize for the confusion. There are two different networks in the study domain: 1) 40+39 WEGN stations and 2) an Austrian national weather station from the network of the national meteorological service (ZAMG). This ZAMG station is a member of the GPCC network. We did not include the station for our study since its data directly affect the performance of IMERG Final run.*

*We have re-written the sentence to "attributed to an Austrian national station, which is not part of the WEGN gauges, but located within the WEGN area" (Page 8 Lines 12-13), so that readers can understand it clearly.*

**3 Figure 5 (the second and third columns) I may miss something, but the WEGN PDFs and CDFs exist for the entire rain rate regions although they are presented for low (1.2mm) rain amounts each. For example, shouldn't the PDFs and CDFs start from 1.2 mm in the third column (high rain rate)? Why do the PDFs and CDFs exist for R<1.2mm?**

*=> Your understanding is correct. WEGN PDFs and CDFs have values only from/till 1.2 mm, but the previous plots contain lines extending beyond stated range because, for example, WEGN PDFc in the high rain rate cases (third column) starts from 0% at 0.05 mm (not at 1.2 mm) so we get to see values even in < 1.2 mm ranges. We now re-plotted the second and third columns so the starting/end points of WEGN PDFs and CDFs are indicated by a vertical line and the WENG PDFc/CDFv end/begin at the line (Page 18, Fig.5). Thank you for the question.*

**4 Time shift in Figures 7 and 8 It is hard to say that the observed patterns in Figures 7 and 8 show a time shift. I think that we can say there is a time shift only when the rainfall durations are the same between reference (WEGN) and IMERG products and starting times are different. It seems to me like that the observed patterns are just errors, probably by morphing and other reasons. In Figure 8, the shapes and peak times are all different and it is hard to find any consistent or systematic tendency.**

*=> Thank you for pointing this out. The term "shift" was used to refer to the time disagreement between WEGN and IMERG; to be clearer, we have re-written the sentence (Page 8 Lines 25-26); "with a time difference of about 2 hours earlier in starting time. This false alarm suggests"*

**Minor comments:**

**1 Page 3 Line 1. ". . .. has been suggested to guarantee a monthly error of under 10%". This cannot be directly applied to this study because of the temporal scale difference (monthly vs. 30-min). The gauge representativeness is a function of space and time scales used (Seo and Krajewski 2010).**

*=> We have added the VRF value obtained from WEGN 30-min data in the  (Page 3 Lines 3-5) as you suggested, please also refer to Major comment #1.*

**2 Page 4 Line 24. Please explain "high native time resolution."**

*=> We have modified the sentence as "WEGN gridded data with 5-min native resolution are used." (Page 4 Line 31)*

**3 Page 4 Line 27. Please clarify the rainfall threshold used in this study. In the caption of Figure 2, there is a phrase "≥ 0.1 mm at single station". How big is the tip resolution against the threshold?**

*=> The caption means that we applied 0.1 mm threshold to count the number of WEGN gauges (the tip resolution is 0.1 mm) that detected rainfall. So this threshold is used only in Figure 2. On the other hand, 0.05 mm threshold is the one used to define rain/no-rain data in each grid box (average of multiple gauges) for the study. We have modified the Figure 2 caption (Page 16) to say "(gauges with >= 0.1mm rainfall are counted)" to make it clearer.*

**4 Page 6 Line 33. Please clarify the difference between Figures 4 and 5. Is the threshold 0.05 mm applied to both Figures 4 and 5?**

*=> Yes, the same threshold is applied. Figure 4 shows all rain data of IMERG and WEGN (i.e. not only hits but also false alarms and misses), whereas Figure 5 shows only matched data (i.e., only hits. This is explained at Page 7 Lines 11-12).*

**5 Page 7 Line 5. Isn't it 0.05 instead of 0.5?**

*=> It is 0.5. We are comparing the POD score of IMERG-WEGN for all rain intensity ranges versus higher intensities (for WEGN > 0.5) to show that misses of IMERG are observed at relatively low rain intensities. To make it clear, we have re-written the sentence of Page 7 Lines 15-19: "Indeed, we recomputed the POD score using only values when WEGN is above 0.5 mm*

*30-min$^{-1}$ (i.e., disregarding low rain rates) and found that the PODs are 0.70, 0.79, and 0.75 for IMERG-E, IMERG-L, and IMERG-F estimates, respectively,". Please also refer to the next comment – Minor comment #6.*

**6 Page 7. Line 6. Please explain "the entire WEGN data." Is it without applying the threshold?**

> *=> It is with applying the threshold. Here "the entire WEGN data" meant all available WEGN data; we corrected the sentence as ", while against WEGN of the entire range of rain intensities above the 0.05 mm 30-min$^{-1}$ threshold, the PODs are only 0.50, 0.57, and 0.53, respectively." in order to make it clear.*

**7 Page 7 Line 31. It would be useful if the correlation coefficient values obtained from other evaluation studies are provided and compared with those of this study.**

> *=> It seems to us that it will be very difficult to directly compare our correlation coefficient values with those of other studies, because the agreement score between ground reference and IMERG can be very different depending on i) the nature of the ground reference (e.g., radar vs gauge, number of gauges, rainfall regime) and on ii) temporal and spatial scales used for comparison. Moreover, given that the study aims to compare the performance between three different IMERG runs, we show the indices in Table 2 to explore differences between the IMERG runs rather than to give absolute scores of each run.*

**8 Page 9 Line 2-4. What (more PMW estimates or rain gauge correction) has more contribution to the improvement? I think that this is very important point in the satellite product evaluation**

> *=> Thank you for the suggestion. We have added "From these two case studies, it appears that the gauges provide a greater improvement to IMERG Final estimates"; Page 9 Lines 18-19.*

**9 Section 4.3. Please add some implication of the result found in Section 4.3 and discussions on how to use the products for hydrologic applications.**

> *=> We are grateful for this comment. We added "This analysis identifies possible sources of error that should be considered in the context of hydrological applications of IMERG data. For instance, biases (overestimation in this case) in IMERG rainfall estimates will inevitably propagate through hydrologic models, and consequently this would lead to larger errors in runoff. The magnitude of biases can be reduced when IMERG Final estimates are used. Time offset bias, however, remains relatively stable across all three IMERG runs, especially in the warm season. Therefore, comparison or adjustment of IMERG estimates using local ground reference (if available) in terms of biases not only in amount of rainfall but also in its timing should be considered as an approach to meet the required level of accuracy in rain data." in <Section 4.3> (Page 10, Lines 4-10)*

***Response to the comments of Dr. Uijlenhoet (Referee #3)
on* "Evaluation of GPM IMERG Early, Late, and Final
rainfall estimates with WegenerNet gauge data in
southeast Austria"
*by* Sungmin O et al.**

*We would like to thank you for the positive feedback on our manuscript. The paper recommended was
helpful to provide an example of recent study on the GPM IMERG (Page 2 Line 16) and also to support
our discussion on the effect of gauge adjustment (Page 9 Line18). Regarding the hydrological
evaluation, some runoff modeling studies with IMERG have been done in the U.S., China, and actually
other locations around the globe, but, we agree that this would be a worthy endeavor for Austrian
catchments.*

[revised manuscript text omitted]

*** Second and third columns of Figure 5. are modified. The starting/end points of WEGN PDFc and CDFv are indicated by a vertical line ant the WEGN PDFc/CDFv end/begin at the line. Please see Major Comment #3 of Referee #2 for details.

[Figure]

**Figure 5.** Same layout per panel as Fig. 4, but restricted to the data pairs for which IMERG and WEGN both detected rainfall ($\geq$ 0.05 mm 30-min$^{-1}$). Comparisons are shown for IMERG-F (top), IMERG-L (middle), and IMERG-E (bottom), with using the entire data (left), the data divided into low and high rain amounts (middle-left, middle) based on the 80$^{\text{th}}$ percentile of WEGN data, or the data divided into two different seasons (middle-right, right), warm (April, May, October) and hot (June to September).

[Figure]

**Figure 6.** Scatter plots of half-hourly rainfall amounts for IMERG-F (top), IMERG-L (middle), and IMERG-E (bottom) versus WEGN, for the entire dataset (left), warm season only (middle), and hot season only (right). Overplotted are, for nine bins across the range with at least 30 IMERG-WEGN data pairs in each bin, the $50^{th}$ percentile (black symbols and line) and the $25^{th}$ and $75^{th}$ percentiles (grey symbols and lines) of the IMERG estimates. Total number of data pairs (n) is indicated in the upper left of each panel. Note that a log-log scale is used.

**Table 2.** Validation statistics comparing the performance of the three IMERG datasets for the two selected grid cells covered by the WEGN.

|  |  | RB | MAE (mm) | NMAE | RMSE (mm) | NRMSE | r | $\rho$ |
|---|---|---|---|---|---|---|---|---|
| IMERG-F | Grid 15.85 | 0.22 | 1.18 | 0.97 | 2.19 | 1.79 | 0.34 | 0.46 |
|  | Grid 15.95 | 0.30 | 1.16 | 0.99 | 2.23 | 1.91 | 0.31 | 0.47 |
| IMERG-L | Grid 15.85 | 0.33 | 1.28 | 1.12 | 2.74 | 2.38 | 0.40 | 0.42 |
|  | Grid 15.95 | 0.54 | 1.41 | 1.29 | 3.31 | 3.02 | 0.30 | 0.44 |
| IMERG-E | Grid 15.85 | 0.28 | 1.27 | 1.11 | 2.84 | 2.47 | 0.38 | 0.38 |
|  | Grid 15.95 | 0.67 | 1.48 | 1.42 | 3.31 | 3.16 | 0.34 | 0.38 |

[Figure]

**Figure 7.** Example rainfall event in the warm season that occurred on 30 May 2015. (left) Spatial rainfall pattern over the WEGN network area for a sequence of four half-hourly WEGN rain rate maps, with the two selected IMERG grid cells indicated (red-framed boxes). (right) Time series of IMERG-F (red), IMERG-L (blue), IMERG-E (green) estimates and WEGN (black) data for the two IMERG grid cells over the period of the rainfall event, with the shaded areas highlighting the two hours illustrated by the map sequence to the left. Solid vertial lines indicate time steps where all three IMERG runs received PMW-based information for the rainfall retrieval at that time step, dotted vertical lines (not applicable for this event but for the event of Fig. 8 below) mean that either IMERG-E or both IMERG NRT runs did not receive PMW-based information at the time step, and no vertical line (the case for most time steps) implies that none of the IMERG runs received PMW-based information.

[Figure]

**Figure 8.** Same as Fig. 7, but for an example rainfall event in the hot season that occurred on 08 July 2015.

[Figure]

**Figure 9.** Contour plots of RMSE $(\mathrm{mm\,30\text{-}min^{-1}})$ between IMERG estimates and WEGN data as a function of WEGN gauge rainfall accumulation time and time offset (which means the time from which to start the WEGN gauge rainfall accumulation relative to the IMERG data start time).

---

## Author Response (AR2)

**Dear Dr. Wouter Buytaert,**

*We appreciate your taking the time to review our manuscript. We also thank both reviewers for their positive feedback. Following the comments from Reviewer 1, we have modified Fig. 7 and Fig. 8 (Page 20 in the marked-up manuscript) and the sentences in Conclusions (Page 10, Lines 23-24).*

*Kind regards,*

*Sungmin O*

[revised manuscript text omitted]